# *BACH1* Expression Is Promoted by Tank Binding Kinase 1 (*TBK1*) in Pancreatic Cancer Cells to Increase Iron and Reduce the Expression of E-Cadherin

**DOI:** 10.3390/antiox11081460

**Published:** 2022-07-27

**Authors:** Liang Liu, Mitsuyo Matsumoto, Miki Matsui-Watanabe, Kyoko Ochiai, Bert K. K. Callens, Long Chi Nguyen, Yushi Kozuki, Miho Tanaka, Hironari Nishizawa, Kazuhiko Igarashi

**Affiliations:** 1Department of Biochemistry, Tohoku University Graduate School of Medicine, Sendai 980-8575, Japan; dongfangweiming2017@yahoo.co.jp (L.L.); m-matsumoto@med.tohoku.ac.jp (M.M.); kochiai@med.tohoku.ac.jp (K.O.); bert.kk.callens@gmail.com (B.K.K.C.); yukohzukiwan@gmail.com (Y.K.); miho.tanaka.t6@dc.tohoku.ac.jp (M.T.); hnishizawa@med.tohoku.ac.jp (H.N.); 2Center for Regulatory Epigenome and Diseases, Tohoku University Graduate School of Medicine, Sendai 980-8575, Japan; 3Department of Neurochemistry, Tohoku University Graduate School of Medicine, Sendai 980-8575, Japan; watanabe-washio@med.tohoku.ac.jp; 4Faculty of Health, Medicine and Life Sciences, Maastricht University, 6200 Maastricht, The Netherlands; 5Ben May Department for Cancer Research, University of Chicago, Chicago, IL 60637, USA; nguyenchilong@uchicago.edu

**Keywords:** *BACH1*, *TBK1*, iron, ferritin, pancreatic cancer

## Abstract

BTB and CNC homology 1 (*BACH1*) represses the expression of genes involved in the metabolism of iron, heme and reactive oxygen species and promotes metastasis of various cancers including pancreatic ductal adenocarcinoma (PDAC). However, it is not clear how *BACH1* is regulated in PDAC cells. Knockdown of Tank binding kinase 1 (*TBK1*) led to reductions of *BACH1* mRNA and protein amounts in AsPC−1 human PDAC cells. Gene expression analysis of PDAC cells with knockdown of *TBK1* or *BACH1* suggested the involvement of *TBK1* and *BACH1* in the regulation of iron homeostasis. Ferritin mRNA and proteins were both increased upon *BACH1* knockdown in AsPC−1 cells. Flow cytometry analysis showed that AsPC−1 cells with *BACH1* knockout or knockdown contained lower labile iron than control cells, suggesting that *BACH1* increased labile iron by repressing the expression of ferritin genes. We further found that the expression of E-cadherin was upregulated upon the chelation of intracellular iron content. These results suggest that the *TBK1*-*BACH1* pathway promotes cancer cell metastasis by increasing labile iron within cells.

## 1. Introduction

Iron is essential for various cellular processes, including cell proliferation and growth [1,2]. It plays a crucial role in DNA synthesis as a cofactor of DNA polymerase and ribonucleotide reductase [3,4,5,6]. Many of the enzymes for oxidation-reduction reactions, including those in the electron transfer chain in the mitochondria and demethylases of DNA, RNA and histone, are dependent on iron, i.e., as iron-sulfur clusters or heme [7,8,9,10,11,12,13]. Iron is also involved in the regulation of diverse proteins including tumor suppressor p53 [14,15]. However, labile ferrous iron, which is not tightly bound to proteins, is toxic to cells, tissues and the body through the generation of reactive oxygen species (ROS) via the Fenton reaction [16,17]. Iron-dependent accumulation of phospholipid hydroperoxide also can cause ferroptosis, an iron-driven cell death [18,19]. To balance the essential and detrimental roles of iron, cellular iron availability is tightly regulated by a network of genes that maintain cellular iron acquisition, storage, utilization and export. Among these, ferritin protects cells from iron-induced oxidative stress and allows for rapid adaptation to changing intracellular concentrations and demand for iron [20,21]. Importantly, cancer cells are known to be more dependent on iron than normal cells [22,23,24,25]. Transferrin receptor 1 is required for iron import by endocytosis, and lipocalin 2 is involved in an alternative pathway of iron uptake; both are highly expressed in many cancers [26,27,28,29], including pancreatic cancer [30,31]. The decrease of ferritin increases the labile iron pool on cells expressing the HRAS oncogene [32,33]. Ferroportin, as the only known iron efflux pump in vertebrates, is downregulated in breast cancer cell lines and prostate cancer and associated with increased levels of the labile iron pool [34,35].

The transcription repressor *BACH1* [36] regulates the expression of genes involved in the metabolism of iron, heme and ROS. For example, *BACH1* represses the expression of ferritin heavy (*FTH1*), light chain genes (*FTL*) and the iron exporter ferroportin (*SLC40A1*) to increase intracellular labile iron [37,38,39]. *BACH1* also represses the expression of heme oxygenase-1 gene (*HMOX1*), which degrades heme and recycles iron [40,41,42]. The repressor activity of *BACH1* is negated by the direct binding of heme to *BACH1* [43]. Therefore, one of the key functions of *BACH1* is the regulation of iron homeostasis by sensing heme. *BACH1* also promotes the malignant properties of cancer cells [43], including metastasis in breast cancer [44,45], pancreatic ductal adenocarcinoma (PDAC) [46] and lung cancer [47,48]. *BACH1* inhibits the expression of genes involved in epithelial cell adhesion, including E-cadherin (*CDH1*), claudin (*CLDN3* and *CLDN4*) and occludin (*OCLN*), to promote epithelial-mesenchymal transition (EMT) and metastasis of PDAC cells [46]. While *CLDN3* and *CLDN4* genes are directly repressed by *BACH1*, *CDH1* is not a direct target gene of *BACH1*. It remains unclear how *BACH1* inhibits the expression of the E-cadherin gene [46]. Exploring the regulation of the E-cadherin gene by *BACH1* may be beneficial to elucidating the pathological roles of *BACH1* in cancer proliferation and metastasis, as well as in the search for new therapeutic approaches.

Tank binding kinase 1 (*TBK1*) is a serine/threonine kinase with an important role in multiple signaling pathways [49,50]. The majority of research on *TBK1* has focused on its role in innate immunity. cGAS-STING pathway activates *TBK1*, which then activates transcription factors IRF3 and NF-κB, resulting in the production of antiviral and proinflammatory cytokines, including type I interferons [51,52,53,54,55,56]. Recently, the role of *TBK1* has been expanded into cancers and autophagy [57,58,59,60,61,62,63,64,65]. *TBK1* has also been reported to promote autophagy-mediated degradation of ferritin to increase intracellular iron [66]. However, it remains unclear whether *TBK1* regulates iron homeostasis via additional mechanisms. Prompted by this finding, here, the relationship between *TBK1* and *BACH1* in the regulation of iron metabolism and E-cadherin gene expression in PDAC cells is studied.

## 2. Methods

### 2.1. Reagents

Deferasirox (DFX) was transferred as raw material from Novartis Pharma (Basel, Switzerland). Dimethyl sulfoxide (DMSO) was purchased from Sigma-Aldrich (St. Louis, MO, USA).

### 2.2. Cells and Cell Culture

Human embryonic kidney 293T (HEK293T) cells were maintained in DMEM-low glucose supplemented with 10% heat-inactivated FBS (Sigma Aldrich, St. Louis, MO, USA), 100 unit/mL penicillin and 100 µg/mL streptomycin (Gibco, Carlsbad, CA, USA). Human pancreatic cancer cell lines AsPC−1 were obtained from ATCC. AsPC−1 was cultured in RPMI-1640 medium supplemented with 20% heat-inactivated FBS, 100 unit/mL penicillin and 100 µg/mL streptomycin (Gibco). SW1990 was cultured in DMEM (Sigma Aldrich), with 10% heat-inactivated FBS, 100 unit/mL penicillin, 100 μg/mL streptomycin (Gibco) and 10 mM HEPES (Gibco, Carlsbad, CA, USA). The cells used were limited to less than 20 passages. *BACH1* knockout AsPC−1 cells were reported previously [46].

### 2.3. Western Blotting

Cells were washed with PBS and harvested. They were then centrifuged at 2300× *g* for 1 min and the supernatant was discarded. The cells were lysed using RIPA buffer (150 mM NaCl, 50 mM Tris-HCl, 1% NP40, 0.5% sodium deoxycholate, 0.1% SDS) for 30 min on ice. Precipitated proteins were removed by centrifugation at 20,400× *g* for 30 min. Protein concentrations were measured using the Pierce 660 nm protein assay (Thermo Fisher Scientific, San Jose, CA, USA) and adjusted.

The supernatant containing the proteins was mixed with SDS sample buffer (62.5 mM Tris-HCl pH 6.8, 1% SDS, 10% glycerol, 1% 2-mercaptoethanol and 0.02% bromophenol blue) containing protease inhibitors (0469315900, cOmplete^®^ Mini EDTA-free Protease Inhibitor Cocktail Tablets, Roche, Mannheim, Germany) and fractionated on slab gels (7.5% or 10% acrylamide separating gel; 4% stacking gel). The gels were run at 100 V for about 2 h and wet-transferred onto polyvinylidene difluoride (PVDF) at 300 mA for 1.5 h at 4 °C. The blots were washed with TBS-T (25 mm Tris, 137 mm NaCl, 3 mm KCl, 0.05% Tween-20, pH 7.4) and incubated for 1 h in TBS-T/5% skimmed milk. The blots were incubated overnight at 4 °C with the primary antibodies diluted in TBS-T containing 5% skimmed milk. On the next day, the blots were washed 3 times for 10 min in TBS-T and incubated for 1 h at room temperature with horseradish peroxidase (HRP)-labeled secondary antibodies diluted in TBS-T containing 5% skimmed milk. The bands were detected using a Clarity Western ECL substrate (Bio-Rad, Hercules, CA, USA) in the ChemiDoc Touch imaging system (Bio-Rad, Hercules, CA, USA) after washing three times for 10 min with TBS-T.

### 2.4. Antibodies

Anti-*BACH1* mAb (1:500, clone 9D11, generated in-house) and anti-*BACH1* antiserum (1:1000, A1–6, generated in-house) were reported previously [46,67]. Other antibodies were ACTB (1:1000, GTX109639, GeneTeX, Irvine, CA, USA), E-cadherin (1:1000, ab1416, Abcam, Cambridge, MA, USA), ferritin light chain (1:100, sc-74513, Santa Cruz Biotechnology, Cambridge, CA, USA), ferritin heavy chain (1:100, sc-376594, Santa Cruz Biotechnology, Santa Cruz, CA, USA), *TBK1* (1:1000, D1B4, Cell Signaling, Boston, MA, USA) and *FBXO22* (1:1000, 13606-1-AP, Proteintech, Rosemont, IL, USA). Anti-rabbit IgG-HRP (1:2500, NA934V, GE Healthcare, Fairfield, CT, USA) and anti-mouse IgG-HRP (1:2500, NA931V, GE Healthcare, Fairfield, CT, USA) were used as secondary antibodies.

### 2.5. siRNAs

Target-specific siRNAs (Stealth RNAi siRNA Duplex Oligoribonucleotides, Invitrogen) were transfected using Lipofectamine RNAiMAX (Thermo Fisher Scientific, Waltham, MA, USA). Stealth RNAi siRNA Negative Control, Low GC (Thermo Fisher Scientific, Waltham, MA, USA) was used as a control siRNA. All siRNA nucleotide sequences are listed in Table 1.

### 2.6. Quantitative Real-Time PCR

RNA was isolated via RNeasy Plus Mini Kit (Qiagen, Valencia, CA, USA) and 500 ng of total RNA was reverse transcribed to single-stranded cDNA using the High-Capacity cDNA Archive Kit (Applied Biosystems, Foster City, CA, USA). Quantitative PCR has performed with LightCycler Fast Start DNA Master SYBR Green I (Roche, Basel, Switzerland) in LightCycler 96 instrument (Roche, Basel, Switzerland). Compared to ACTB, we found that the expression of *RPL13A* was more stable when DFX was used. After referring to other papers [68,69], we chose *RPL13A* as the housekeeper gene. All primer sequences are listed in Table 2.

### 2.7. Immunofluorescent Staining

Cells were fixed in 4% formaldehyde/PBS. After incubation with anti-*BACH1* (1:200, 9D11) antibodies for 1 h at 37 °C, the antigen-antibody complexes were detected by an anti-mouse IgG FITC-conjugated secondary antibody (1:1000, F3008, Sigma Aldrich, Darmstadt, Germany). Hoechst 33258 (Thermo Fisher, Waltham, MA, USA) was used at 20 µg/mL to stain the nuclei.

### 2.8. Detection of Labile Iron and Cell Death

The level of Fe^2+^ was measured using 5 µM Mito-FerroGreen (Dojindo Molecular Technologies, Tokyo, Japan) according to the manufacturer’s protocol. 4′,6-diamidino-2-phenylindole (DAPI) was used for the assessment of cell death. AsPC−1 cells were sorted with a FACS Verse (BD) and analyzed by FlowJo software (Tree Star). The gating strategy for living cells is shown in Appendix A.

### 2.9. RNA Sequence

First, 4 µg of the total RNA was purified using an RNeasy Plus Mini Kit. Host cell rRNA contamination was removed using the GeneRead rRNA Depletion Kit (QIAGEN, Hilden, Germany) and RNA was purified using an RNeasy MiniElute Cleanup Kit (QIAGEN, Hilden, Germany). Next, 100 ng rRNA-depleted RNA was randomly sheared at 95 °C for 10 min. The sample was purified by a Magnetic Beads Cleanup Module (Thermo Fisher Scientific, Carlsbad, CA, USA). The genomic libraries were constructed using an RNA-seq library kit ver. 2 (Thermo Fisher Scientific) in the AB Library Builder system (Thermo Fisher Scientific). Each library was barcoded with Ion Xpress RNA Seq-Barcode 01-16 Kit (Thermo Fisher) to enable multiplex sequencing. Amplified segments were then size-selected (100–200 bp) using Agencourt AMPure XP magnetic beads (Beckman Coulter, Brea, CA, USA). Sequencing templates were prepared on an Ion Chef System using the Ion PI Hi-Q Chef Kit (Thermo Fisher Scientific). Samples were sequenced on the Ion Proton System using the Ion PI™ Hi-Q™ Sequencing 200 Kit and Ion PI™ v3 chip. Raw data were obtained as fastq files. The sequence data were aligned to reference hg19 by the Ion Torrent RNASeqAnalysis plugin (Thermo Fisher Scientific). Mapped reads were counted using htseq-count v 0.9.1 and performed likelihood ratio test by using the edgeR package v 3.16.5 after the removal of low-read-count genes (count per million <5). RNA-seq data of *siTBK1* from this study have been deposited in GEO under SuperSeries accession number GSE201307.

### 2.10. The Analysis and the Visualization of Public Data

The dataset used comprised mRNA-seq data from TCGA tumors (see TCGA Data Portal at https://tcga-data.nci.nih.gov/tcga/, accessed on 17 October 2021). The two-gene correlation map is realized by the R software package ggstatsplot (3.3.3). We used Spearman’s correlation analysis to describe the correlation between quantitative variables without normal distribution. A p-value of less than 0.05 was considered statistically significant.

Data from GSE124408 in Gene Expression Omnibus (GEO, https://www.ncbi.nlm.nih.gov/geo, accessed on 9 December 2021) was used as the ChIP-seq data of *BACH1* in AsPC−1 and SW1990. DNA sequences of genes are listed in Table 3. PDF files were made ‘igvtools ver. 2.3.93′ and visualized using Integrative Genomics Viewer [70,71].

### 2.11. Statistics

Statistical analyses were performed by GraphPad Prism 8. For all experiments, differences in data sets were considered statistically significant when *p*-values were lower than 0.05. When comparing only two groups, an unpaired Student’s *t*-test was performed. To compare multiple groups, one-way ANOVA was used. *, *p* < 0.05; **, *p* < 0.01, ns (not significant) *p* > 0.05.

## 3. Results

### 3.1. TBK1 Promotes BACH1 Expression and Accumulation

We analyzed the mRNA expression data from TCGA using the GEPIA database (Gene Expression Profiling Interactive Analysis) [72] and found that the expression of *BACH1* and *TBK1* are positively correlated in pancreatic cancer (Figure 1A). Since pancreatic cancer is a cancer of the digestive system, we also analyzed several other cancers of the digestive system and found similar results in bowel cancer, hepatocellular carcinoma and gastric cancer (Appendix A). To determine the roles of *TBK1*-mediated regulation of *BACH1* in PDAC cells, we knocked down *TBK1* by siRNA in human AsPC−1 PDAC cells. *TBK1* knockdown led to the reduction of *BACH1* protein and mRNA levels (Figure 1B). Because the change of the mRNA was smaller than that of the protein, a major regulatory mechanism of *BACH1* by *TBK1* may be at the protein level rather than the transcription level in AsPC−1 cells. Phosphorylation of *BACH1* by *TBK1* (L.L. et al., submitted) also suggested this possibility. In contrast, when *TBK1* was knocked down in HEK293T cells, endogenous *BACH1* protein and mRNA levels did not decrease (Appendix A). To test if *TBK1* affects the subcellular distribution of *BACH1*, immunostaining was carried out. *TBK1* knockdown resulted in a decrease of the endogenous *BACH1* protein in AsPC−1 cells, but its subcellular distribution did not change (Appendix A). The cell counts did not change under the reduction of *TBK1* (Appendix A). These results indicated that *TBK1* is required for the accumulation of *BACH1* protein and to promote *BACH1* function in PDAC cells.

### 3.2. TBK1-BACH1 Pathway Regulates Iron Homeostasis and Cell Migration

To further investigate the regulatory relationship between *TBK1* and *BACH1*, we knocked down *TBK1* in AsPC−1 cells and carried out an RNA-seq analysis. Using our previously published data of RNA-seq in AsPC−1 cells with or without *BACH1* knockdown [46], there were 102 common upregulated genes and 181 common downregulated genes in *TBK1*-silenced and *BACH1*-silenced AsPC−1 cells (Figure 2A,B). A GO pathway enrichment analysis of these genes showed that terms related to oxidation-reduction process, regulation of transcription from RNA polymerase II promoter in response to iron, positive regulation of sequence-specific DNA binding transcription factor activity and cellular iron ion homeostasis were enriched in genes upregulated in response to *TBK1* or *BACH1* knockdown (Figure 2C). Considering that *TBK1* is involved in iron metabolism [66], these results suggested the involvement of *TBK1* in the regulation of *BACH1* to tune iron homeostasis in AsPC−1 cells. Consistent with the reduction of *BACH1* protein, its known target genes were increased in their expression, including *HMOX1* and *SLC40A1*. On the other hand, the enrichment of terms related to positive regulation of endothelial cell proliferation, wound healing and positive regulation of cell migration in the genes whose expression was decreased (Figure 2D). The expression of mesenchymal genes, such as *MMP7*, *SNAI2*, and *VIM*, were decreased upon *TBK1* knockdown (Appendix A). The expression of these genes is also reduced upon *BACH1* knockdown in these cells [46], suggesting that *TBK1* promotes metastatic process including cell migration via enhancing the function of *BACH1*.

### 3.3. BACH1 Increases the Iron Content by Reducing the Expression of Ferritin

A reanalysis of ChIP-seq data of *BACH1* in AsPC−1 cells [46] showed direct binding of *BACH1* to *FTL* and *FTH1* encoding ferritin subunits (Figure 3A). Ferritin mRNAs were increased upon knockout (Figure 3B) or knockdown (Appendix A) of *BACH1* in AsPC−1 cells. Correspondingly, ferritin heavy and light chain proteins were increased in AsPC−1 cells with the knockout of *BACH1* (Figure 3C). While *BACH1* also directly binds with *SLC40A1* encoding ferroportin, its mRNA was not increased (Appendix A). Two cytoplasmic iron regulatory proteins (*IRP1* and *IRP2*) post-transcriptionally regulate cellular iron metabolism, including translational inhibition of ferritin mRNAs when the iron is reduced [20,73,74]. *IRP2*, but not *IRP1*, was also decreased under the situation of *BACH1* knockdown and knockout (Appendix A), suggesting a direct or indirect regulation of *IRP2* expression by *BACH1*.

To examine whether these alterations led to changes in iron within cells, we measured labile iron, which is not bound tightly to proteins, in mitochondria by using Mito-FerroGreen and flow cytometry (FACS) with a gating strategy shown in Appendix A. FACS analysis showed that AsPC−1 cells with *BACH1* knockout and knockdown were lower in the mean intensity of the Mito-FerroGreen signal and less in the mean numbers of Mito-FerroGreen positive cells than respective control cells (Figure 3D; Appendix A). These results suggested that *BACH1* increases mitochondrial labile iron in AsPC−1 cells by repressing the expression of ferritin genes.

### 3.4. Iron Availability Regulates BACH1 and E-Cadherin Expression

Since iron and heme play key roles in the regulation of *BACH1* [43,75,76,77], the changes in *BACH1* protein and mRNA in response to fluctuations of iron in AsPC−1 and SW1990 derived from PDAC patients was examined with ferrous sulfate (FeSO_4_) or iron chelator deferasirox (DFX). *BACH1* protein was increased with DFX and decreased with FeSO_4_ (Figure 4A,B; Appendix A). Consistent with the protein amount, *BACH1* mRNA increased in response to DFX after 12 or 48 h. Among downstream target genes of *BACH1*, *HMOX1* mRNA decreased with DFX (Figure 4C; Appendix A). Taken together with the enhanced *BACH1* protein degradation by heme [77], these results demonstrated that iron deficiency, which leads to a decrease in heme, induces the expression of *BACH1* at both transcriptional and post-translational steps.

*BACH1* is polyubiquitinated by the E3 ligase adaptor proteins F-box protein 22 (*FBXO22*) [47]. The mRNA of *FBXO22* decreased with DFX (Figure 4C; Appendix A), which suggested that *FBXO22*-dependent *BACH1* degradation was decreased under this condition, contributing to the accumulation of *BACH1* protein. The next question is how *TBK1* responds to a reduction of iron. *TBK1* mRNA and protein were decreased when AsPC−1 cells were treated with DFX (Figure 4C,D; Appendix A). This observation led to the question of whether *TBK1* is involved in the accumulation of *BACH1* protein upon iron chelation. DFX still increased *BACH1* protein even when *TBK1* was reduced by siRNA (Figure 4D). However, the accumulation of *BACH1* was reduced compared with DFX alone. *FBXO22* protein was also reduced by DFX, which was not affected by *TBK1* knockdown (Figure 4D). When taken together, these observations suggested that *TBK1* and iron constitute two independent, parallel pathways to control *BACH1* expression in PDAC cells. *TBK1* promotes the accumulation of *BACH1* protein whereas iron inhibits its accumulation.

While *BACH1* reduces the expression of *CDH1* encoding E-cadherin in PDAC cells to promote metastasis [46], the specific mechanism remains unclear. A recent report pointed to the role of iron in promoting EMT in HepG2 cells [78]. When taken with this report, the above observations raise the possibility that the regulation of *CDH1* expression involves iron. Indeed, E-cadherin mRNA and protein were both increased by the DFX treatment (Figure 5A). The increased expression of E-cadherin upon *BACH1* knockdown may involve iron reduction caused by increased expression of ferritin (Figure 5B).

## 4. Discussion

This study has shown that *TBK1* is required to maintain the expression of *BACH1* in PDAC cells. Since *TBK1* phosphorylates *BACH1* (LL et al., unpublished observation), this modification may increase the stability thereof. In addition, *TBK1* was found to increase *BACH1* mRNA. Since *TBK1* is known to activate transcription factors for immune responses such as IRF3 and NF-κB [51,52,53,54,55,56], these factors may transactivate *BACH1* gene expression. As iron metabolism is dynamically altered during infection [79,80,81], *TBK1* and *BACH1* may contribute to immune responses as well as EMT by altering iron metabolism. The links between iron and E-cadherin expression remain unclear. It has been reported that in colon cancer cells, iron chelators antagonize the reduction of E-cadherin expression in response to transforming growth factor-β, and that their effect is dependent on the induction of N-myc downstream-regulated gene 1 (*NDRG1*) upon iron chelation [82]. Therefore, it will be important to examine whether *NDRG1* expression is altered in response to *TBK1* or *BACH1*. *TBK1* is known to promote EMT, including the reduction of E-cadherin expression, by altering the activities of downstream protein kinases such as AKT, ERK and GSK3β [83,84]. However, it was also reported that *TBK1* inhibits EMT in breast cancer cells via increasing the expression of estrogen receptors [85]. Taken together with these reports, our findings suggested that *TBK1* may promote EMT and metastasis of cancer cells via both *BACH1*-independent and *BACH1*-dependent mechanisms. Consistent with this hypothesis, it has been reported that *TBK1* enhances the invasive and metastatic capacity of pancreatic cancer cells [86].

The expression of *BACH1* in PDAC cells is strictly regulated in response to the amount of intracellular iron. As a continuously high level of *BACH1* is expected to increase cellular iron levels, which may cause ferroptosis [87], an alternative mechanism to suppress overshooting of *BACH1* appears to be necessary. Our results suggest that this may be achieved by the decrease of *BACH1* by iron, leading to the re-expression of ferritin genes. When iron is limited, cells increase the amount of *BACH1* by two distinct mechanisms. One is the increased transcription of *BACH1* mRNA in response to a reduction in iron. Since *BACH1* is known to be induced by the hypoxic response [88], hypoxia-inducible factors may be involved in the increased transcription of *BACH1* gene in response to iron deficiency. The mechanism of the increased expression of *BACH1* mRNA awaits further clarification. The other mechanism involves a decrease in the expression of *FBXO22*, which leads to a cessation of *BACH1* degradation. As heme is a complex of iron and protoporphyrin IX, the supply of iron via the transferrin-receptor pathway limits, and thus controls, the heme synthesis rate [89]. Hence, it is also possible that DFX enhances *BACH1* accumulation by reducing heme-regulated degradation of *BACH1* [77]. The presence of multiple pathways to increase *BACH1* in response to iron reduction probably leads to a rapid increase in *BACH1* protein level, resulting in prompt changes in target gene expression.

The present observations suggest that *BACH1* and iron form a negative feedback loop (Figure 5B). When it is active, *BACH1* increases intracellular labile iron by repressing the expression of ferritin genes, and *TBK1* is required to maintain the amount of *BACH1*. Ensuring an increase in mobile iron and heme then decreases the amount of *BACH1* protein in part by increasing the expression of *FBXO22* and promoting heme-dependent degradation of *BACH1*. Therefore, iron and *BACH1* regulate each other by a network of multiple regulatory interactions including *TBK1* and *FBXO22*. It will be important to investigate the universality and cell type-specific modulation of this network in various cell types including cancer cells.

## 5. Conclusions

We determined that *BACH1* protein and mRNA levels were reduced by *TBK1* knockdown in PDAC cells. A GO pathway enrichment analysis of PDAC cells with knockdown of *TBK1* or *BACH1* suggested that *TBK1* and *BACH1* both regulate cellular iron ion homeostasis and cell migration. Consistent with this, *BACH1* represses the transcription of ferritin and thus, brings about an increase in cellular iron levels in these cells. In addition, iron reduces the expression of E-cadherin. These observations suggest that *TBK1* and *BACH1* promote metastasis of PDAC cells, in part by controlling iron metabolism.

## Figures and Tables

**Figure 1 antioxidants-11-01460-f001:**
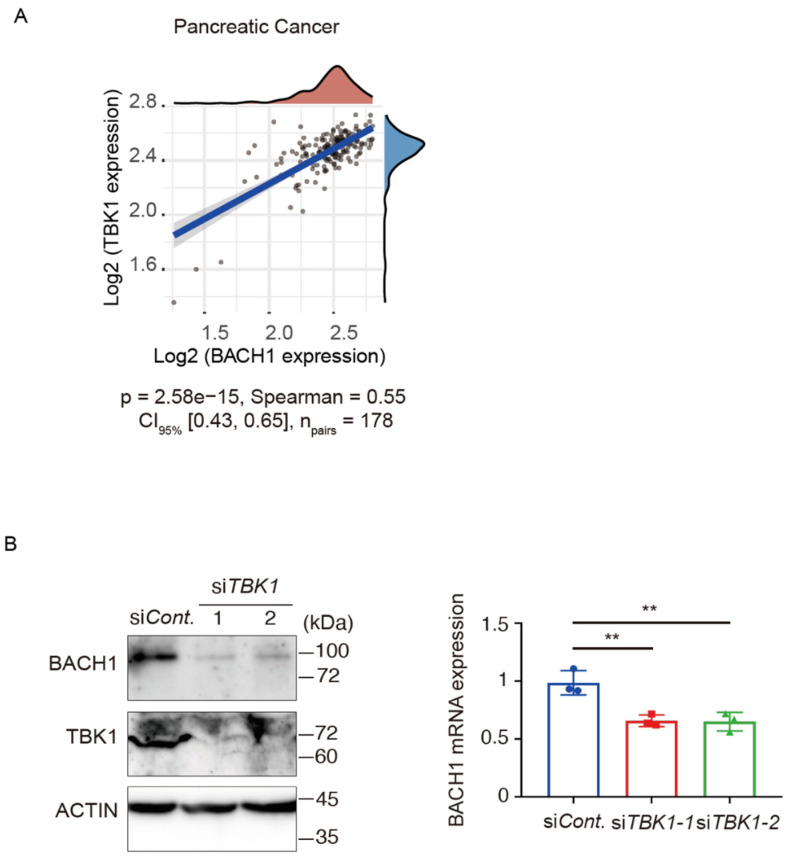
*TBK1* promotes *BACH1* expression and accumulation. (**A**) Spearman correlation analysis of *BACH1* and *TBK1* gene expression in 178 pairs pancreatic cancer samples. The horizontal and ordinate axes in the figure represent the expression distribution of *BACH1* and *TBK1*, respectively, with density curves representing their distributions. p represents correlation *p* values, Spearman means Spearman’s rank correlation coefficient, CI_95%_ shows 95% confidence limits. (**B**) Effects of *TBK1* knockdown on *BACH1*. At 48 h after *TBK1* knockdown in AsPC−1 cells, cell lysates were used for anti-*BACH1*, anti-*TBK1* and anti-actin Western blotting (left) or measurement of *BACH1* mRNA levels (right). Scrambled siRNA was used as a negative control. One-way ANOVA, *n* = 3 biologic replicates for each experiment. **, *p* < 0.01.

**Figure 2 antioxidants-11-01460-f002:**
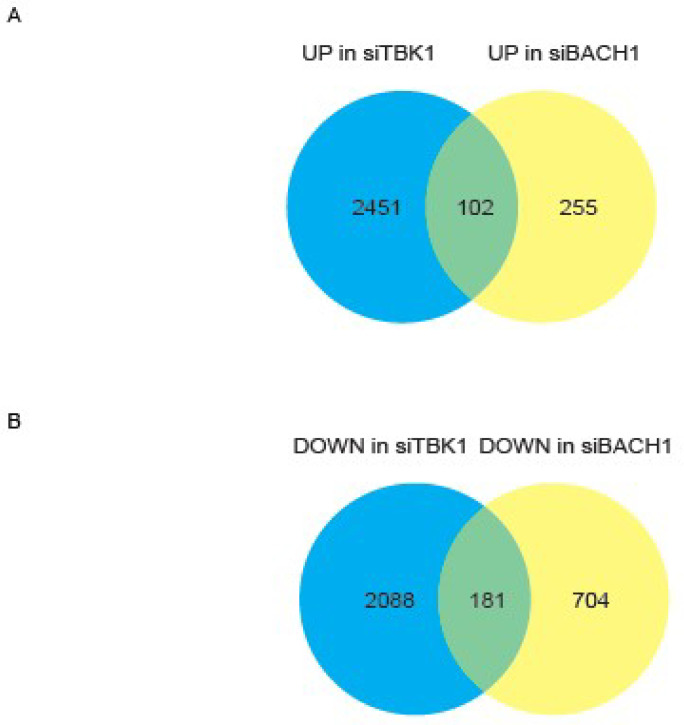
*TBK1* and *BACH1* regulate genes for iron homeostasis and cell migration. (**A**,**B**), AsPC−1 cells with *TBK1* siRNA or control siRNA were analyzed with RNA-sequence, resulting in upregulated (**A**) or downregulated (**B**) genes with *TBK1* knockdown. These genes were compared with those affected with *BACH1* knockdown, pointing to 102 common upregulated genes and 181 common downregulated genes in *TBK1*-silenced and *BACH1*-silenced AsPC−1 cells. (**C**,**D**), Top 10 Gene Ontology (GO) biological process terms enriched in the upregulated genes (**C**) or downregulated genes (**D**) using DAVID for each considered group.

**Figure 3 antioxidants-11-01460-f003:**
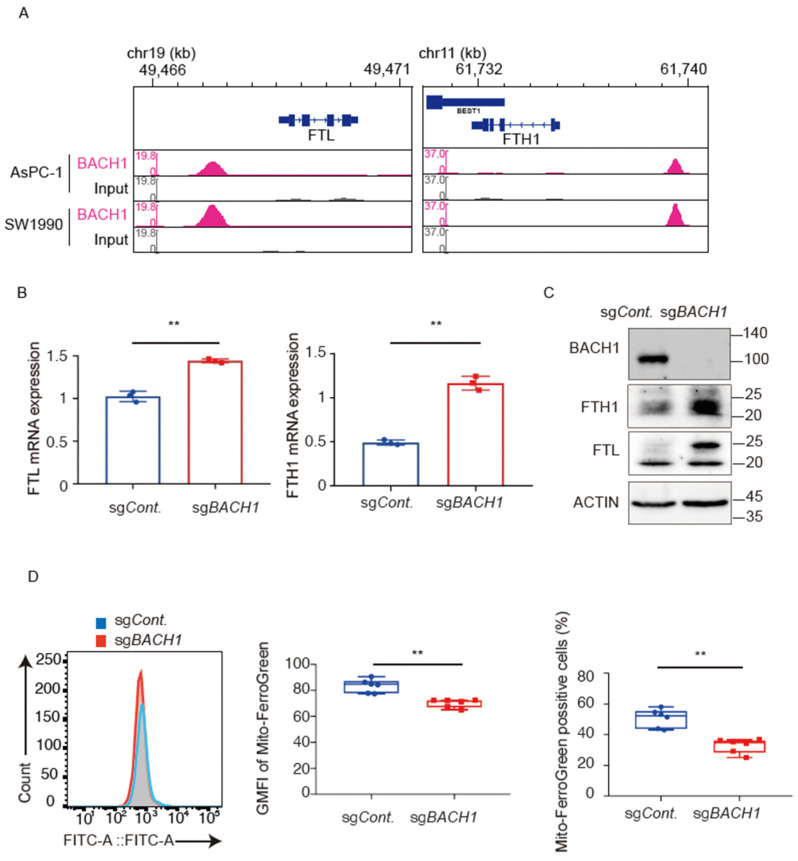
*BACH1* increases the iron content by repressing the expression of ferritin. (**A**), ChIP-seq analysis of the binding of *BACH1* to *FTL* and *FTH1* genes in AsPC−1 cells and SW1990 cells. (**B**), Relative mRNA levels of *FTL* and *FTH1* in *BACH1* knockout and control AsPC−1 cells. All data are presented as mean ± SD, with *p* values from the Student’s *t*-test, *n* = 3 biologic replicates for each experiment. **, *p* < 0.01. (**C**), Western blotting of *BACH1* and ferritin proteins as in (**B**). Cell lysates were used for anti-*BACH1*, anti-ferritin heavy, anti-ferritin light and anti-actin Western blotting. (**D**), Flow cytometry analysis for detecting mitochondrial Fe^2+^ with Mito-FerroGreen in *BACH1* knockout and control AsPC−1 cells. Distribution (left), mean fluorescence intensity (middle) and the fraction of positive cells (right) are shown. GMFI, geometric mean fluorescent intensity. All data are presented as mean ± SD, with *P* values from the Student’s *t*-test. *n* = 3 biologic replicates for each experiment. **, *p* < 0.01.

**Figure 4 antioxidants-11-01460-f004:**
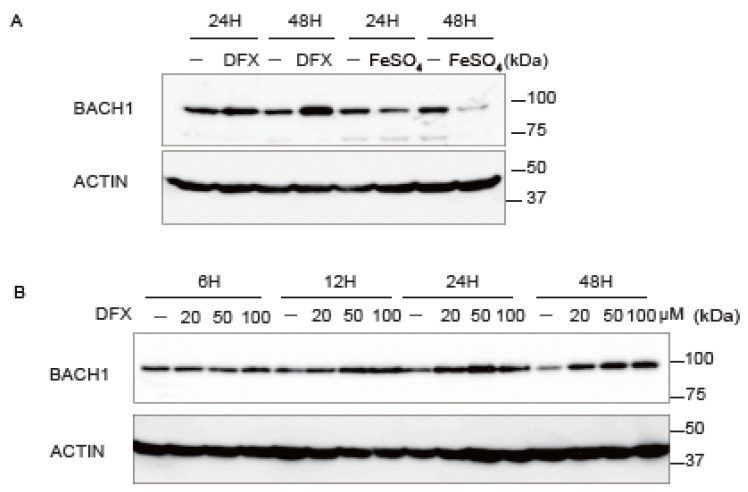
The increases of *BACH1* mRNA and protein under iron deficiency. (**A**), Western blotting of *BACH1* protein. AsPC−1 cells were treated with DFX (100 μM) or FeSO_4_ (200 μM) for 24 h or 48 h and cell lysates were used for Western blotting with anti-*BACH1* and anti-actin antibodies. DMSO served as a negative control. (**B**), Western blotting of *BACH1* protein. AsPC−1 cells were incubated with indicated concentrations of DFX for 6, 12, 24 or 48 h. Proteins were detected as in (**A**). (**C**), Relative mRNA levels of indicated genes. AsPC−1 cells were incubated with DFX (50 μM) or DMSO for 12 h. mRNA amounts were normalized using actin mRNA and presented as mean ± SD, with *p* values from the Student’s *t*-test. *n* = 3 biologic replicates for each experiment, *, *p* < 0.05; **, *p* < 0.01. (**D**), Western blotting of *BACH1*, *TBK1* and *FBXO22* proteins. At 24 h after *TBK1* knockdown, cells were incubated with DFX (50 μM) or DMSO for 48 h and were used for Western blotting. DMSO served as a negative control.

**Figure 5 antioxidants-11-01460-f005:**
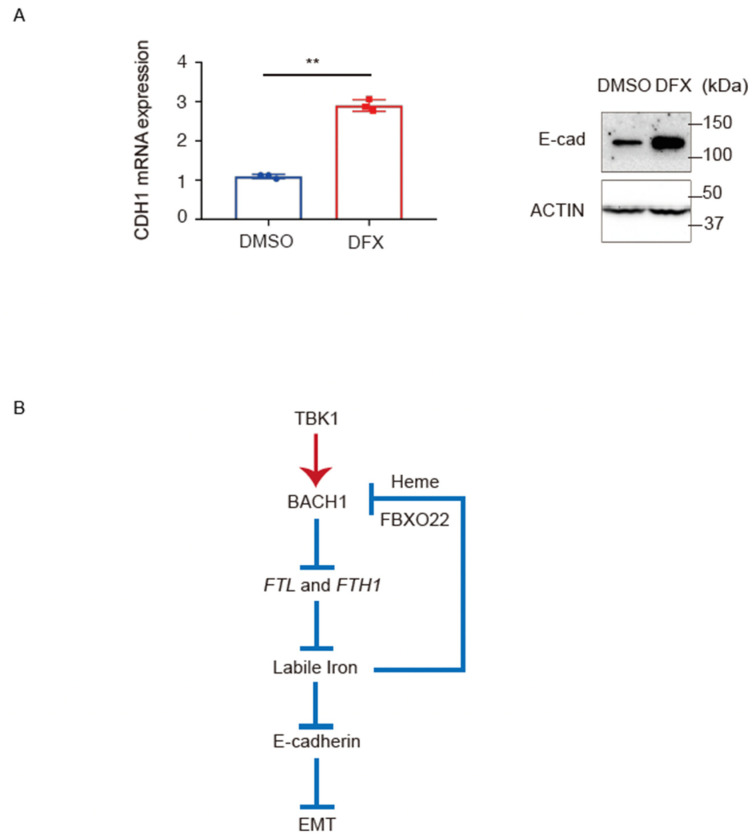
The increases of E-cadherin mRNA and protein under iron deficiency. (**A**), *CDH1* mRNA (left) and protein (right) levels. AsPC−1 cells were incubated with DFX (50 μM) or DMSO for 48 h. All data are presented as mean ± SD, with *p* values from the Student’s *t*-test. *n* = 3 biologic replicates for each experiment. **, *p* < 0.01. (**B**), Proposed model for the regulation of iron and *BACH1* and their connection to the regulation of E-cadherin in pancreatic cancer cells.

**Table 1 antioxidants-11-01460-t001:** Sequences of siRNAs.

siRNA Name	Accession Name	Sequence (5′-3′)
*siBACH1-1*	NCBI Gene ID 571	GGUCAAAGGACUUUCACAACAUUAA
*siBACH1-2*	NCBI Gene ID 571	GGGCACCAGGGAAGAUAGUAGUGUU
*siTBK1-1*	NCBI Gene ID 29110	GGACUACCAGAAUCUGAAUUCUUAA
*siTBK1-2*	NCBI Gene ID 29110	GCGAGAUGUGGUGGGUGGAAUGAAU
*siTBK1-3*	NCBI Gene ID 29110	GGGAACCUCUGAAUACCAUAGGAUU

**Table 2 antioxidants-11-01460-t002:** Sequences of qPCR primers.

Gene Name	Accession Name	Forward Primer (5′-3′)	Reverse Primer (5′-3′)
*BACH1*	NCBI Gene ID 571	GTTACTTCCACTCAAGAATCGT	ACATTTGCACACTTCATCCA
*CDH1*	NCBI Gene ID 999	TCCTGGCCTCAGAAGACAGA	CCTTGGCCAGTGATGCTGTA
*FBXO22*	NCBI Gene ID 26263	ATTGCTTGGTTCGCGTGGTA	GCTCTCTTATGGCCACGACA
*FTH1*	NCBI Gene ID 2495	TGAAGCTGCAGAACCAACGAGG	GCACACTCCATTGCATTCAGCC
*FTL*	NCBI Gene ID 2512	TACGAGCGTCTCCTGAAGATGC	GGTTCAGCTTTTTCTCCAGGGC
*HMOX1*	NCBI Gene ID 3162	TTTCAGAAGGGCCAGGTGAC	AGTAGACAGGGGCGAAGACT
*IRP1*	NCBI Gene ID 48	TGCTTCCTCAGGTGATTGGCTACA	TAGCTCGGTCAGCAATGGACAACT
*IRP2*	NCBI Gene ID 3658	ACCAGAGGTGGTTGGATGTGAGTT	ACTCCTACTTGCCTGAGGTGCTTT
*MMP7*	NCBI Gene ID 4316	ATCATGATTGGCTTTGCGCG	CCAGCGTTCATCCTCATCGA
*RPL13A*	NCBI Gene ID 23521	TCGTACGCTGTGAAGGCATC	GTGGGGCAGCATACCTCG
*SLC40A1*	NCBI Gene ID 30061	GATCCTTGGCCGACTACCTG	CACATCCGATCTCCCCAAGT
*SNAI2*	NCBI Gene ID 6591	CAACGCCTCCAAAAAGCCAA	ACAGTGATGGGGCTGTATGC
*TBK1*	NCBI Gene ID 29110	AGCGGCAGAGTTAGGTGAAA	CCAGTGATCCACCTGGAGAT
*VIM*	NCBI Gene ID 7431	GGACCAGCTAACCAACGACA	GGGTGTTTTCGGCTTCCTCT

**Table 3 antioxidants-11-01460-t003:** DNA sequences of genes analyzed.

Gene	NCBI Reference Sequence
*BACH1*	NG_029658.2
*CDH1*	NG_008021.1
*FTH1*	NG_008346.1
*FTL*	NG_008152.1
*SLC40A1*	NG_009027.1
*TBK1*	NG_046906.1

## Data Availability

RNA-seq data of *siTBK1* from this study have been deposited in GEO under SuperSeries accession number GSE201307.

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
