# Peer review of "BACH1 Expression Is Promoted by Tank Binding Kinase 1 (TBK1) in Pancreatic Cancer Cells to Increase Iron and Reduce the Expression of E-Cadherin"

_antioxidants, 2022, doi:10.3390/antiox11081460_

Round 1
Reviewer 1 Report
Ferritin gene repression by BACH1 in pancreatic cancer cells depends on Tank 2 binding kinase 1 (TBK1) and promotes the iron-mediated reduction of E-cadherin 3 expression
GENERAL COMMENTS
The work reported in this manuscript was conceived as a follow-on from a previous article published article (Nishizawa et al 2019) and the other was submitted for publication (L.L. et al, submitted) as stated by the authors. The report evaluated the Tank 2 binding kinase 1 (TBK1) regulation of BACH to repress ferritin expression in pancreatic cancer cells. RNA-seq, flow cytometry and gene silencing, ontology were used to gather evidence to conclude that expression of E-cadherin was up-regulated upon chelation of intracellular iron content. In general, the style of writing could be better and devoid of too many personal pronouns.
A list is presented below of suggestions on areas requiring the authors’ attention for the amendment to the text.
Minor points
2/45 ---- ‘However, it remains unclear how cancer cells maintain a higher amount of intracellular iron’. The literature abounds on the molecular mechanisms of processes and the regulation of iron transport proteins that account for this.
2/72---- ‘We have recently found that TBK1 phosphorylates BACH1 at multiple serine and threonine residues (L.L. et al, submitted)’. The current manuscript seems premature and could be submitted after the publication of L.L. et al,
3/77----define all abbreviations at first stating eg DFX, BX795
5/148---- A FAC analysis with Mito-FerroGreen was employed to determine the labile iron, a more robust quantitative method of analysis is imperative to substantiate the involvement of iron in the physiological phenomenon being proposed.
5/174----Public data seems vague
Overall, the methods are not described in sufficient detail and the presentation is haphazard without any explanation or narration of the purpose of the experimentation process. Some other cell lines were used without any justification for why they were chosen.
7/209 ---- TBK1 expression correlates with BACH ??
Main points
Although the study confirmed the evidence that BACH1 increases the expression of ferritin and E-cadherin mRNA and protein are increased during iron deficiency, attempts at the involvement of BACH in the regulation of E-cadherin are preliminary. Fig 5 is a proposal but the title of the manuscript is affirmative, the mechanistic link of the regulation of iron and BACH1 and E-cadherin is not supported by the data currently.
Consequently, the title of the manuscript requires rephrasing since these assertations are currently putative. L.L. et al,’s publication will lend credence to the message of the manuscript. Pleiotropy of genes, CHP-seq associations of proteins and ontology annotations of genes are complex and caution should be exerted to define the functions of pertinent proteins while avoiding possible confounders in systemic metabolism. The discussion seems too suggestive, speculative and further explanation of the molecular processes is required.
Author Response
Our replies and responses are in the Word file.

Reviewer 2 Report
Ferritin gene repression by BACH1 in pancreatic cancer cells depends on Tank binding kinase 1 (TBK1) and promotes the iron-mediated reduction of E-cadherin expression
Liu et al present an interesting investigation into the regulation of E-cadherin expression in pancreatic cancer; possibly defining a new therapeutic target.
The paper is well written and for the most part, data well presented.
I think the study would be improved with the inclusion of some simple experiments investigating the effect of these treatments /knock downs on cell growth or migration. Even if these show no impact, I think the effects at a cellular level are important to report and strengthen the conclusions that the authors can make.
Minor points.
Fig 1b; the blot is not great quality. This is a very important figure for the paper, so I think if possible a better quality blot which is not cropped so tightly would be preferable.
Give full name of DFX in the methods
Comment on the choice of housekeeper gene for RT_PCR analysis- are the authors sure the treatment conditions do not influence the expression of this gene.
Author Response

(The authors gave the same response as above.)
